# Performance and Long Distance Data Acquisition via LoRa Technology of a Tubular Plant Microbial Fuel Cell Located in a Paddy Field in West Kalimantan, Indonesia

**DOI:** 10.3390/s19214647

**Published:** 2019-10-25

**Authors:** Emilius Sudirjo, Pim de Jager, Cees J.N. Buisman, David P.B.T.B. Strik

**Affiliations:** 1Environmental Technology, Wageningen University & Research; Bornse Weilanden 9, 6708WG Wageningen, The Netherlands; pim.dejager@wur.nl (P.d.J.); ; (C.J.N.B.); 2Government of Landak Regency, West Kalimantan Province, 79357 Ngabang, Indonesia; 3Plant-e BV, Mansholtlaan 4, 6708PA Wageningen, The Netherlands

**Keywords:** tubular, plant, microbial fuel cell, electricity, rice, paddy fields, LoRa, bioelectrochemical system

## Abstract

A Plant Microbial Fuel Cell (Plant-MFCs) has been studied both in the lab and in a field. So far, field studies were limited to a more conventional Plant-MFC design, which submerges the anode in the soil and places the cathode above the soil surface. However, for a large scale application a tubular Plant-MFC is considered more practical since it needs no topsoil excavation. In this study, 1 m length tubular design Plant-MFC was installed in triplicate in a paddy field located in West Kalimantan, Indonesia. The Plant-MFC reactors were operated for four growing seasons. The rice paddy was grown in a standard cultivation process without any additional treatment due to the reactor instalation. An online data acquisition using LoRa technology was developed to investigate the performance of the tubular Plant-MFC over the final whole rice paddy growing season. Overall, the four crop seasons, the Plant-MFC installation did not show a complete detrimental negative effect on rice paddy growth. Based on continuous data analysis during the fourth crop season, a continuous electricity generation was achieved during a wet period in the crop season. Electricity generation dynamics were observed before, during and after the wet periods that were explained by paddy field management. A maximum daily average density from the triplicate Plant-MFCs reached 9.6 mW/m^2^ plant growth area. In one crop season, 9.5–15 Wh/m^2^ electricity can be continuously generated at an average of 0.4 ± 0.1 mW per meter tube. The Plant-MFC also shows a potential to be used as a bio sensor, e.g., rain event indicator, during a dry period between the crop seasons.

## 1. Introduction

Plant microbial fuel cells (Plant-MFC) have drawn the attention of many researchers since their first proof of principle performance using Reed mannagrass (*Glyceria maxima*) [1]. In principle the Plant-MFC is based on the microbial fuel cell (MFC) in which electrochemically active bacteria (EAB) generate electrons from substrates (i.e., glucose or acetate) at the anode; the generated electrons are transferred to the anode electrode and flow via an external load to the cathode side. At the cathode, these electrons react with a final electron acceptor (i.e., oxygen) and protons [2]. The electricity production depends on the anode and the cathode potential differences, in which the cell potential is the cathode potential minus the anode potential [2,3,4]. The common approach to estimate the theoretical cell potential is using thermodynamics of the anode (e.g., acetate) and the cathode (e.g., oxygen) reaction calculated based on Gibbs free energy at a specific condition [2,3]. For instance, the acetate oxidation (C_2_H_3_O_2_^−^ + 4 H_2_O→2 HCO_3_^−^ + 9 H^+^ + 8 e^−^) and oxygen reduction (O_2_ + 4 H^+^ + 4 e^−^→2 H_2_O) potential at a specific condition (acetate concentration 0.05M, [H_2_O] = 1M; pH = 7; pO = 0.2 bar; T = 298 K) against Ag/AgCl reference electrode are −0.494V and 0.6V, respectively [3]. Therefore, a theoretical MFC cell potential using acetate model substrate and oxygen as the final electron acceptor is 1.094V. In addition, the Plant-MFC has growing plants which provide substrates in form of rhizodeposits that warrant electron donor availability for electrochemically active bacteria (EAB) to generate electricity [1].

Plant-MFCs have been studied both in the lab [1,5,6,7,8,9,10,11,12,13,14,15,16,17] as well in field studies [18,19,20,21,22,23]. So far, the field studies were performed using a conventional plant-MFC design which puts the anode in a lower anaerobic part (below the surface of the soil) and the cathode above the soil surface to obtain oxygen [18]. For a large scale application, this design is less practical because one needs to excavate the top layer of soil before anode installation. To avoid this excavation, Timmers proposed a new tubular Plant-MFC design which could be installed using a horizontal drilling technique [24]. Plant-MFCs’ can potentially be applied in wet agricultural lands because they do not tend to disturb the surface that in this case can still be used for food production. Once implemented in for example, a rice paddy, a challenge is the (re-)use of the Plant-MFC over sequential rice cultivation cycles. Rice paddies are typically manually and/or mechanically managed via various activities: (a): preparation of land by ploughing; (b) seedling and transplanting of the rice paddy; (c) water management; (d) fertilization; (e) weeding, and (f) pest control. This rice paddy management should not destroy the Plant-MFC so it can be used for many years. Also the rice plant growth can potentially negatively affect the Plant-MFC performance by e.g., root growth into the cathode electrode which would provide rhizodeposits at the wrong electrode. Another potential negative effect is the excretion of oxygen at the anode, causing inhibition due to cathodic reduction reactions at the anode [8]. Earlier tubular plant-MFC’s were evaluated during long-term lab-based peat soil wetlands tests reaching a two week average power generation of 21 mWm^−2^ plant growth area (PGA) [17].

Wetlands are considered the suitable place to integrate Plant-MFC’s technology in a real life application because a Plant-MFC requires anaerobic conditions for the anode (to minimise competing oxygen reduction processes) and allow plants growth to supply rhizodeposits and other potential organic substrates to generate electricity (like organic plant matter remains after plants harvesting or leaf littering) [25]. The total amount of paddy fields represent one of largest types of wetlands. In 2017, the average harvested area of paddy fields and the rice paddy production in the world were about 167 million ha and 770 million tonnes of rice, respectively [26]. As the human population is growing, the need for paddy fields and production to support basic human need for food is also growing. More than 90% of world rice paddies are located in Asia [26]. With their current size in the world, paddy fields have a great potential to be integrated with plant-MFC technology. Plant-MFC also has potential to reduce emissions of methane, a well-known potent green-house-gas, from paddy fields [6,27].

Even though there were several studies concerning Plant-MFCs in the field, to our best knowledge no study has been performed yet on the performance of tubular Plant-MFCs in a paddy field. The objective of this study was to investigate the performance of a tubular plant-MFC in a tropical paddy field, providing data analysis over a whole rice paddy growing cycle. Electricity generation, rice growth and the microbial community were studied. An online data acquisition method using a long range (LoRa) technology was developed since local manual measurements on electricity generation could not be done at sufficient intervals [28,29,30]. The paddy field area had no connection to the electricity grid and there were no long-range transmissions network available. Therefore, a local LoRa network was set-up. The end devices using 2 Volt input (2 × AA alkaline batteries) were connected to the Plant-MFC to measure voltages and temperature. A LoRa gateway was installed and locally powered by a solar power system. This gateway provided the communication with the data acquisition devices via LoRa connection and sent the collected data to a LoRa network server through an IP connection (3G internet) [28]. The set-up and performance of this long distance data acquisition and transfer from West Kalimantan, Indonesia to Wageningen, The Netherlands, is also discussed. 

## 2. Materials and Methods

### 2.1. Paddy Field

An existing paddy field in West Kalimantan, Indonesia (0.919215 N, 109.468182 E; elevation 100 m above sea level) was used for the experiments. This paddy field was cultivated using a common agricultural practice in that area. The rice paddy (*Oriza sativa* L. var Inpari 30 Ciherang Sub 1) was transplanted twice per year which are in May and November (Table 1). The rice paddy growth phases are shown in Figure 1. Prior to transplantation, the rice paddy seeds were grown in a nursery site for 20–30 days. Afterward, the rice paddy were manually transplanted by farmers in the soil in lines with a distance between 15 cm and 25 cm. During the seedling period, soil was prepared by ploughing it with a hand tractor. The ploughing was preceded by flowing water into the paddy field (about 7 days) to make the soil softer. The ploughing was done per plot and in line with water flow direction from both up- and downstream. The water was kept flowing between 5–10 cm above the ground in the paddy field until ripening period. The soil was dried in the last 2–3 weeks before harvest. 

Rice paddy maintenance includes several activities such as fertilization, pest control, weeding, and water management. The fertilization was applied twice during one crop season. The first fertilization was about 7 days before transplantation day (TD) after the ploughing and the second fertilization was 7 days after TD. In the first fertilization, three kinds of fertilizer were applied together, which are urea (PT.Pupuk Sriwidjaja, Palembang, Indonesia), NPK Phonska (PT.Petrokimia Gresik, Gresik, Indonesia) and Petroganik (PT.Pupuk Indonesia (PERSERO) Group, Cikampek, Indonesia). In the second fertilization, only urea and NPK Phonska were applied in the paddy field. The application of fertilizer followed the dosage stated on the package. During the fertilization, water inlet and outlet was closed for 2–3 days so that the fertilizers dissolved and seeped into the soil. After that the water was slowly put back into the paddy field with a small flow.

The pest control was done by using molluscicides and insecticides. The molluscicides (Keong Tox, PT.Santani Sejahtera, Medan, Indonesia) was applied 1 day before the first fertilization to control the snail, *Pomacea* spp., (known as keong mas in Indonesian). The insecticides were distributed by spraying a mixture of Mipcindo 50WP (PT. Inti Everspring Indonesia, Mangunreja-Serang, Indonesia) and Imidacloprid 96TC (PT.Catur Agro Dinamika, Pamulang Tanggerang Selatan, Indonesia). The mixing ratio between Mipcindo and Imidacloprid was 3:1 (tablespoon) for 25 L of water. The insecticide spraying was done three times during the crop season which were 12–14 days after TD, 21–25 days after TD and 40 days after TD. In addition, if walang sangit (*Leptocorisa oratorius*) and wereng (a general term used to designate plant-liquid sucking insects from the order *Hemiptera*) were still seen a lot, another insecticide spraying was done (at 60 days after TD) with a 3:2 mixing ratio.

Water management plays an important role in rice paddy cultivation. Before applying the molluscicides water was reduced to about 0.5–1 cm above the ground. This water level was kept until the first fertilization and the transplantation day. Three day after the TD water level was slowly increased to the normal level. The same mechanism was done during the second fertilization. Apart from these period, the water level was kept on the normal level (5–10 cm above the soil) following the rice paddy growth phase. The continuous flooding also functions to reduce weeds. In the studied paddy field, weeds growth was hampered because the fields were well ploughed. Therefore, weeding was only manually extracted if they were spotted. The water flow was stopped 2–3 weeks before harvest day.

The Inpari 30 Ciherang Sub 1 paddy rice is well-known as a flood-tolerant variety. It can tolerate submergence for up to 14 days [32]. According to Indonesian Center for Rice Research, Indonesian Agency for Agricultural Research and Development, Ministry of Agriculture Republic of Indonesia (*Balai Besar Penelitian Tanaman Padi*, *Badan Penelitian dan Pengembangan Pertanian*, *Kementerian Pertanian Republik Indonesia*), this variety has an average productivity of 7.2 ton/ha and the harvest time is 111 day after seedling establishment [33]. 

### 2.2. Electricity Generation

On 28 October 2017, three tubular plant microbial fuel cells (tubular plant-MFCs) were installed in the paddy field (Figure 2). The plant-MFC reactors were installed in lines next to each other from west to east (Appendix A). The distance between reactors was 30 cm. The tubular plant-MFC was manufactured by Plant-e (Wageningen, The Netherlands) according to a similar design used in [17]. As base a transparent silicon tube (VMQ silicone, 12/16 mm inner/outer diameter; rubbermagazijn.nl, Zoetermeer, The Netherlands) was used to supply oxygen. Consequently the cathode felt, the spacer and finally the anode felt was wrapped around this tube. Both the anode and the cathode were made of carbon felt (KFA-5 mm thickness, SGL Carbon GmbH, Bonn, Germany). The electrode length of each tubular plant-MFC was 1 m and the width for the anode and the cathode was 19 cm and 10 cm, respectively. The spacer was made from non-conductive materials to prevent short circuiting, but is otherwise completely permeable (air filter cloth DA/290, DACT Filter- & Milieutechniek, Kerkrade, The Netherlands). Titanium wire (grade 2, 0.5 mm; Titaniumshop BV, Kampen, The Netherlands) was used as current collector both in the anode and the cathode side. The current collector was tied and wrapped around the cathode and the anode as shown in Figure 2.

The tubular plant-MFCs were installed manually by hoeing the top soil of the paddy field prior to the transplantation. The tubular plant-MFCs were placed about 10–15 cm below the soil (Figure 3). Both ends of the silicon tubes were bent down with the open hole facing the ground to avoid rainwater from getting into the tube, which would affect the air, i.e., oxygen, supply to the cathode. 

### 2.3. Measurements and Analysis

Plant-MFC reactor performances were evaluated based on anode potential, cathode potential and cell potential, in combination with different applied external loads. In the first three crop seasons, data were irregularly measured with a digital multimeter (Fluke, Fluke Europe B.V., Eindhoven, The Netherlands). All potentials were measured and reported against a 3M KCl Ag/AgCl reference electrode (QIS, Oosterhout, Netherlands). The anode reference electrode was fixed on the anode surface with a cable tie and the cathode reference electrode was inserted in the tube between the cathode and spacer. In the fourth crop season, data (the anode potential, the cathode potential, the cell potential and temperature) were automatically logged using LoRa technology (AE Sensors B.V, Dordrecht, The Netherlands). The temperatures were measured 50 cm above the ground using the a temperature sensor integrated in the same LoRa data acquisition technology.

Rainfall data were obtained from the two nearest weather station, which are the Mempawah Climatology Station (0.07500 N, 109.19000 E; 2 m above sea level) about 100 km to the south-southwest of the research area and the Paloh Climatology Station (1.74000 N, 109.30000 E; 15 m above sea level) about 100 km north-northwest of the research area. 

Soil samples were collected (on 30 June 2018) from six different locations for microbial community analysis (Figure 4). Samples were grouped into three: Group I (Samples A and C) was soil that attached on the anode from mid plant-MFC; Group II (samples B and D) was soil that attached on the anode from end of the plant-MFC; and Group III (samples E and F) was from soil with 2 m distance from plant-MFC 1 and 2. After collection, samples were kept in a 30 mL-tube container and keep in a 4 °C fridge. The next day samples were transported for 48 h with a cool-ice box for DNA extraction to Genetika Lab, Jakarta (PT. Genetika Science Indonesia, Jakarta, Indonesia), a partner company of 1st BASE Axil Scientific Pte Ltd., Singapore. 

Sequencing steps were performed by 1st BASE [34] as follows: universal primers that targeted the V3V4 regions were used for amplification. The quantity and quality of the PCR product that targeted the V3V4 regions were measured using Tapestation 4200, picogreen and nanodrop. All the samples passed the QC measurement and proceed straight to a library preparation. The libraries were prepared using Illumina 16s metagenomics library prep kit and their quality and quantity were determine using Agilent Tapestation 4200, Picogreen and qPCR. These libraries were then pooled according to the protocol recommended by the Illumina and proceed straight to sequencing using MiSeq platform at 2x301PE format by 1st BASE Axil Scientific Pte Ltd., Singapore.

During the harvest time from the third and the fourth crop season, aboveground biomass was collected from 12 different locations. Three locations (above plant-MFC) were a 1-square-meter area above each plant-MFC reactors and another six location (1 metre from plant-MFC) were a-1-square-metre area both to the north and to the south from each reactor (Appendix A). In the fourth crop season, the biomass was only collected from the north part of the plant-MFC reactors. Biomass was cut about 5 cm from the ground. After collection, wet biomass was weighed using a manual 10-kg counterweights scale with a precision of 100 g.

### 2.4. Long Distance Data Acquisition

Eight wireless voltage meters (AE Sensors, Dordrecht, The Netherlands) were installed for data acquisition on 14 February 2019 between crop season 3 and 4. Each sensor has three inputs; one common ground and two completely differential inputs. For each installed tube, a reference sensor (3M KCl Ag/AgCl reference electrode, QIS, Oosterhout, The Netherlands) was installed which was connected to the common ground. Both the anode and cathode were measured completely differential against the reference input. Each voltage meter is built into a watertight junction box and is powered by two AA batteries, the projected operating time at Borneo conditions with four measurements per hour is at least one year. The voltage meters have a RM186-SM module (LoRa/BLE 868MHz LoRa EU, Laird Connectivity, London, UK) implemented and can be accessed by Bluetooth through the Laird Toolkit app to check it’s status and connectivity (Figure 5). Through the same module, data can be sent through the Long Range Wide Area (LoRaWAN) network. Since the voltage sensors, including the Laird module, were manufactured in The Netherlands, they used the EU LoRaWAN protocol which cannot be used outside of the EU. Moreover, there was no LoRaWAN network installed yet at the research site. We therefore also installed a gateway (Laird RG186 LoRa Gateway, Laird Connectivity) on site that was connected through the locally installed WiFi network dongle (ZTE MIFI Router, InternationalSIM, Terborg, The Netherlands) (Figure 5). This WiFi network was finally made possible through the available 3G mobile phone network (IM3 Ooredoo, PT. Indosat Tbk, Jakarta, Indonesia). Due to the lack of available electricity on site, the whole system is powered by a locally installed battery system (100 Wp solar panel, 100 AH 12 V Rechargeable Sealed Lead Acid Battery; PWM20 Solar Charge Controller) on solar panels. Data is temporary stored by the sensors and sent in CSV-format on a daily basis to pre-defined email addresses of the involved researchers in The Netherlands. This data logging equipment was co-designed and/or manufactured with support by Plant-e B.V (Wageningen, The Netherlands) & AE Sensors B.V (Dordrecht, The Netherlands).

### 2.5. Calculation

Current generation was calculated according to Equation (1):(1)$$I=\frac{V}{R},$$
where *I* is the current production in Ampere (A), *V* is the cell potential in Volt (V) and *R* is the applied load in Ohm (Ω). 

Power generation was calculated according to Equation (2) or Equation (3):(2)P =V×I,
(3)$$P\;=I^{2}\;\times R,$$
where *P* is the power output in Watt (W), *V* is the cell potential in Volt (V), *I* is the current production in Ampere (A) and *R* is the applied load in Ohm (Ω).

Current density (A/m^2^) and Power density (W/m^2^) were obtained by dividing the current production and the power output with plant growth area (PGA). The PGA was 0.0585 m^2^. Energy density (Wh/m^2^) was obtained by multiplying the power density with the time it was generated (h).

Internal resistance (*R_int_*) is calculated according to Equation (4) [4]:(4)$$R_{int}=\frac{E_{OCP}-E_{cell}}{i},$$
where *E_OCP_* is the open cell potential in V, *Ecell* is the measured cell potential in V, *i* is the current density in A/m^2^ and *R_int_* is the internal resistance in Ω·m^2^.

Internal resistance is the accumulation of resistances in the Plant-MFC system due to anode overpotential (ŋ*_an_*), cathode over potential (ŋ*_cath_*) and membrane potential (Em) [4]. Cathode internal resistance (*R_cath_*) and anode internal resistance (*R_an_*) are calculated according to Equations (5) and (6), respectively:(5)$$R_{cath}=\frac{{ŋ}_{cath}}{i}=\frac{E_{OCP,cath}-\;E_{cath}}{i},$$
(6)$$R_{an}=\frac{{ŋ}_{an}}{i}=\frac{E_{an}-\;E_{OCP,\;an}}{i},$$
where *E_OCP_,cath* is the cathode potential at open cell potential (V), *E_OCP_,an* is the anode potential at open cell potential (V), *E_cath_* is the measured cathode potential (V) and Ean is the measured anode potential (V). The internal resistance is reported normalized to the PGA.

In this study, the theoretical cathodic reduction reaction potential of oxygen to water (−0.494V vs. Ag/AgCl) is considered as the cathode potential at open cell condition (*E_OCP_,cath*). The theoretical anodic oxidation reaction of acetate (0.6V vs. Ag/AgCl) is used as the anode potential at open condition (*E_OCP_,an*). Thus, the open cell potential is 1.094V vs. Ag/AgCl [2,3].

The maximum power generation was evaluated by a polarization technique. Polarization was conducted on 29 November 2017. First, the plant-MFCs were operated under open cell conditions (the external load was disconnected) for 10 min. After that the external load was reconnected and changed every 10 min from high to low in order. The external loads used for the polarization were 1000, 470, 220, 100 and 10 Ohm. Cell potential generated from the plant-MFCs for every operated external load were measured with a multimeter after 10 min of operation. From these cell potential, current generation and power generation were calculated according to Equations (1) and (2) and normalized to PGA. Note that this is not an indicator for the actual long term performance of the Plant-MFC since this method does allow to take capacitive current into account.

## 3. Results

### 3.1. Rice Paddy Maintenance, Growth and Production above Plant-MFC Application

The tested Plant-MFCs did generate electricity and the paddy field maintenance was not strongly influenced during this field study. The rice paddy owner took care not to destroy the buried tubes during land preparation and other maintenance activities. Rice plants did grow and produced rice during the experiments (Appendix A). In all four crop seasons, no negative effect was observed in the rice paddy growth due to the tubular plant-MFC installation (Appendix A). Conversely, the aboveground biomass production was on average between 27% and 35% higher above the tubular plant-MFC installation compared to one meter away from that location (Table 2). Although one cannot claim that this increase was due to the plant-MFC installation as there is variation in biomass production possible due to, e.g., the so-called border effect [35]. This result is still important to prove that the co-installation of the Plant-MFC in a paddy field is suitable both for food and electricity generation. At moment of submission of this paper, the Plant-MFCs are still installed and no signs of impairment on the technology were revealed (data not shown). Further continuation could show the durability of the Plant-MFC and provide information on required maintenance. No maintenance was required during the first two years.

### 3.2. Continuous Power Production during the Rice Growing Season

During the rice growing season, a continuous electricity production was achieved (Figure 6A,B and Appendix A). A proper analysis was obtained from the fourth crop season in which data were automatically logged using LoRa technology. Maximum daily average power density of the triplicate experiment reached 9.6 mW/m^2^ PGA (Figure 6A).

Plant-MFC 2 was able to continuously generate a power density of 8.5 mW/m^2^ PGA for 60 days on the fourth growth season (Figure 6A). Based on 72 days of current generation in the fourth crop season during waterlogged conditions, 9.5–15 Wh/m^2^ PGA energy density (at an average of 0.4 ± 0.1 mW per meter tube) was achieved. Maximum of 44 mW/m^2^ PGA power density was achieved using a polarization curve; this maximum power density must be carefully evaluated because it may be influence by a capacitive current [13] (Appendix A). This result was in the same order with other Plant-MFC power density in the paddy field (Table 3). There were several peaks in power density (shown in green circles; best visible at Plant-MFC 1 and 2) as shown in Figure 6A. Such peaks may have different origins that maybe related to the management of the rice paddy as a response to supply of fertilizer or potentially pesticides. They could have an effect on the electrochemically active bacteria in the paddy field. There is no direct evidence to support this last hypothesis, though it is known that pesticides on wetland have effects on rice field microflora [36]. In addition, it could be that the degradation of isoprocarb/mipcin (from Mipcindo 50) and imidacloprid (both chemicals have an aromatic structure like hexaclorobenzene) might provide an additional substrate for the EAB to generate electricity as have been shown that pesticide hexachlorobenzene degradation is enhanced in a soil MFC while generating electricity [37]. Also, it is known that addition of compost can affect current generation in Plant-MFC as well as that specific nutrients could include additional electron donor for electricity generation like in the case for urea obtained from urine [38,39,40].

In the fourth crop season, before the paddy field was irrigated the anode potentials were almost similar to the cathode potential, which were around 500 mV (Figure 6B). At this point, the anode overpotential was higher than the cathode overpotential. Thus, the anode resistance was higher than the cathode resistance (Figure 7).

There were several points that the anode potential went down to around 300–100 mV. These phenomena will be discussed further in Section 3.4. When the paddy fields were watered, the anode potentials gradually decreased up to −369 mV while the cathode potential remained stable at around 400 mV (Figure 6B). This means that the cathode overpotential was relatively stable and the anode overpotential decreased. However, when the irrigation was stopped the anode potential increased rapidly and became higher than the cathode potential (Figure 6B and Figure 7). This indicated a rapid increase in the anode overpotential over the cathode overpotential. Based on this, one can conclude that the cathode is not the limiting factor in this Plant-MFC as can be seen from Figure 7. In addition, the weak peaks in the anode and the cathode potential profile in Figure 7 may be related to diurnal (day and night) effect as described in other studies [15]. However we did not specifically study this effect for this article.

### 3.3. Microbial Community in the Paddy Field

Microbial analysis shows that bacteria dominated the archaea in the paddy field soil. Total relative abundance of archaea was only 1.5 ± 0.8% (Figure 8). In the phyla level, Acidobacteria were the predominant bacteria followed by Proteobacteria, Planctomycetes and Verrucomicrobia. The bacterial communities in the paddy field were diverse (Appendix A). More than 520 genera at least were identified. Proteobacteria were mainly from the classes Betaproteobacteria, Deltaproteobacteria and Alphaproteobacteria. 

In general, the microbial communities from soil attached on the anode and the soil 2 m apart from the anode are relatively similar (Figure 8). However, there is an exception for *Proteobacteria* which were less enriched on the soil attached on the anode of Plant-MFC. This difference is related to the *Comamonadaceae* family (order: *Burkholderiales,* Class: *Betaproteobacteria*). In the class level, Betaproteobacteria relative abundance was slightly higher in the soil far from the plant-MFC (18.5 ± 1%) than in the soil attached on the anode Plant-MFC (10.5 ± 4%). At the order level *Burkholderiales* were more abundant in the soil far from the reactor (9–16%) and less abundant in the soil attached to the anode (0.8–4.7%). This was from the *Comamonadaceae* family, which was found to be more abundant in the soil far from the Plant-MFC (12 ± 4%) than in the soil attached on the anode of the Plant-MFC (1.7 ± 1.3%). However, the genera level of this family cannot be identified. 

The results also show that Deltaproteobacteria are present in all soil samples with a relative abundance of 2.7 ± 0.2%. These bacteria were enriched on the anode of Plant-MFC using rice plants [5]. Some species within the Deltaproteobacteria class that are known to generate electricity are *Geobacter,*
*Deferrisoma,* and *Desulfobulbus* [5].

### 3.4. Plant-MFC Can Indicate a Rain Event as Biosensor

In between the crop seasons, the test paddy field was left in a dry condition. At these moment there was a non-continuous electricity production. During dry conditions the anode potential quickly increased to equal the cathode potential. However, there were several points in this period that the anode potential went down and cell potential peaks were observed (Figure 9). As the Plant-MFC likes anaerobic waterlogged conditions for the anode to generate electricity, these peaks suggested that the paddy field was inundated by water. Since the access of irrigation water to the area was stopped, the only possible reason for this inundation was rain. Since there is no rain data from the research site, rain data from two nearest climatology stations were utilized to predict the rain events at the research site.

Plotting the rain data (Appendix A) and the cell potential peak on the same graph as shown in Figure 9, one can see that there is some correlation between the rain and the cell potential peaks. This result suggests that there is a possibility to utilize Plant-MFC as a bio-indicator (such as of a rain event or as a wetland drying indicator). In Figure 9, there are some rain events (around 27 February 2019 to 6 March 2019) that occurred without being accompanied by cell potential peaks. In addition, there are some peaks occurring (around 30 March 2019 to 5 April 2019) when there was no rain. In both cases, they happened on low rainfall (<20 mm). The first case indicates that the rain state did not reach the research site. Moreover, the second case might indicate there is local rain in the research site. It should be noted that there is a mountain close by the research site. *Bawang* mountain (*Bawang* means onion in Indonesia Language) is situated about 8 km to the west side of the research site. Bawang mountain stretches for 15 km from (0.856579, 109.465752) to (0.960704, 109.360416). The local climate of the research site is also influenced by the presence of this mountain, for instance via orographic rainfall [42]. Therefore, actual rain data from the Plant-MFC installation location are needed to further clarify this correlation. In this study, although it is known that temperature does affect current generation in the lab, temperature is not likely to influence the plant-MFC performances since the temperature in the research site was rather constant during the day and night [43] (Appendix A).

### 3.5. Data Acquisition via Low Range Network Integrated with 3G Network Enable Long Distance Auto Data Collection

Right after installation, the performance of the data acquisition was tested by comparing the anode potential, the cathode potential and the cell potential measured manually with a multitester and result delivered from the LoRa data logger. This step is important for data validation. During this research, there were four instances of data transfer disconnections. However, the connections reconnected again automatically after 2–3 days and only one time there was a need to restart the LoRa gateway. This problem was most likely happened because of the 3G network in the research site was not stable due to weather conditions. Since this research site is located in an area without an electricity grid, it is important to maintain the electricity supply for the system. In addition, enough credit on the mobile phone network was also a key to send the data to the 3G network. Based on this experience, LoRa technology is reliable to be utilized for automatic online long distance data acquisition. 

## 4. Conclusions

Based on this study, we can conclude that installing a plant-MFC in a paddy field is possible and it can generate electricity over several crop cycles. Tubular plant-MFCs can generate electricity continuously during the crop season as long as the rice paddy was flooded. In one crop season, 9.5–15 Wh/m^2^ PGA electricity energy was generated continuously. In between the crop seasons, Plant-MFCs may be utilized as rain event indicators. This opens an opportunity to utilize Plant-MFC technology as a biosensor. Automatic long distance data transfer is possible via LoRa technology.

## 5. Associated Content

Microbiota data (raw 16s rDNA amplicon sequences) was submitted to the EBI database (https://www.ebi.ac.uk/ena) under accession number PRJEB34787.

## Figures and Tables

**Figure 1 sensors-19-04647-f001:**
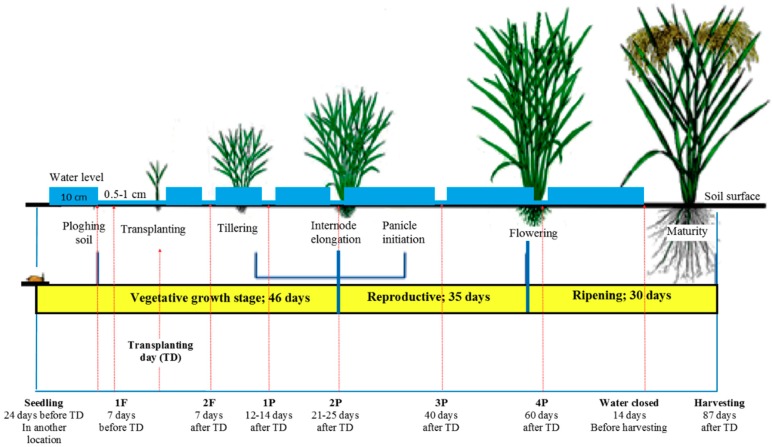
Growth phases of a rice paddy with water, fertilizer and pesticide management. Adapted from [31] under CC BY NC SA3.0 license.

**Figure 2 sensors-19-04647-f002:**
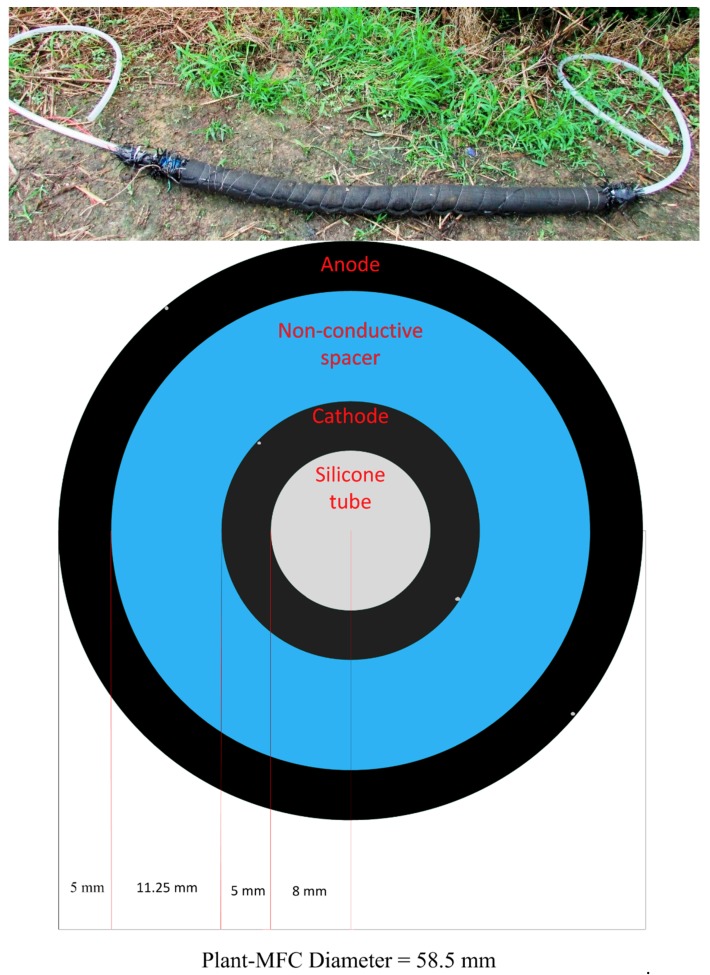
Plant-MFC tubular reactor before installation (above) and its schematic cross-section view (below). The two grey dots on the outer side of the anode and the cathode represent titanium wire current collector.

**Figure 3 sensors-19-04647-f003:**
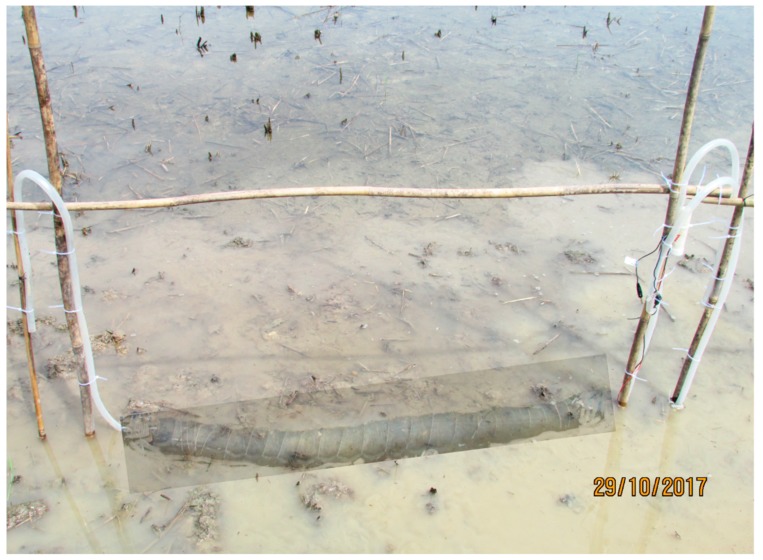
Installed tubular Plant-MFC installation (see shaded area) just before covering the tube with paddy field soil.

**Figure 4 sensors-19-04647-f004:**
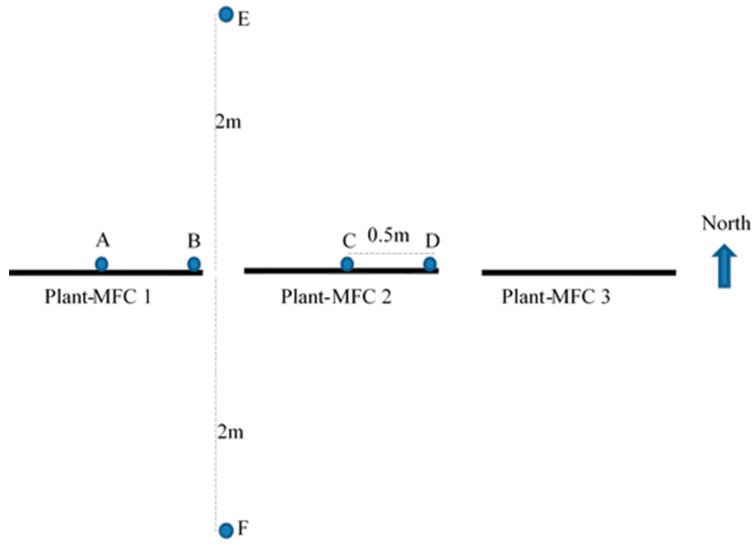
Soil samples collection points.

**Figure 5 sensors-19-04647-f005:**
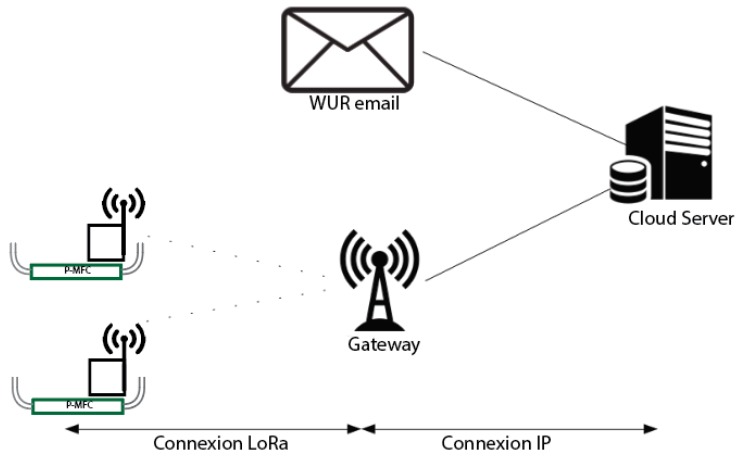
Online data acquisition system.

**Figure 6 sensors-19-04647-f006:**
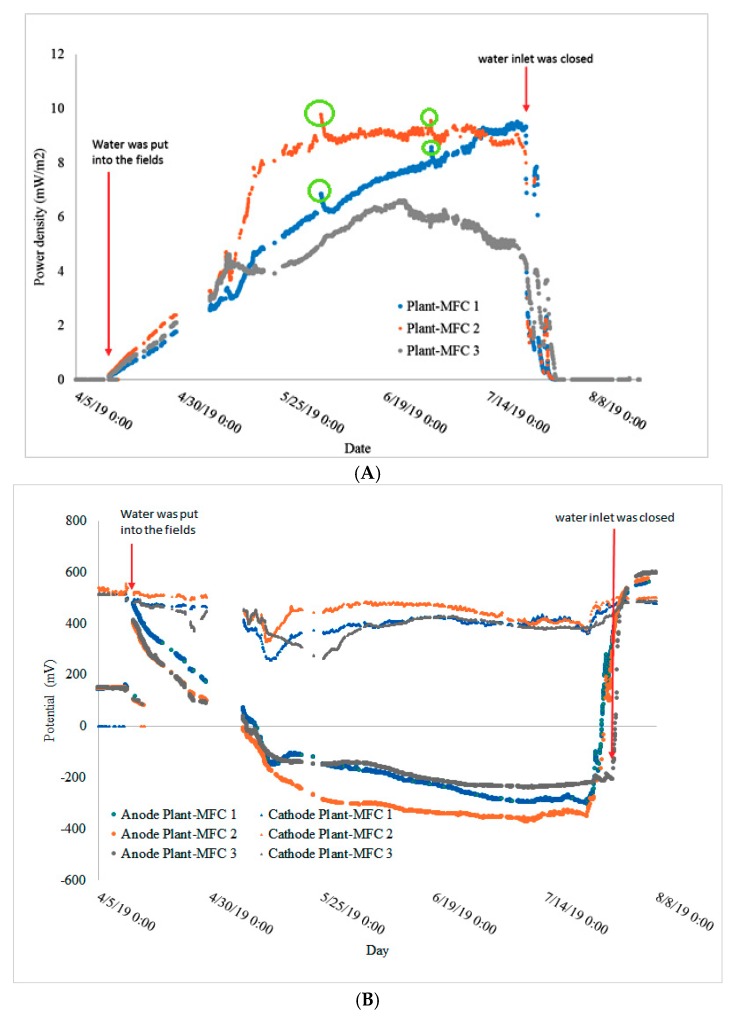
Tubular plant-MFCs performances during fourth crop season: (**A**) power density, and (**B**) anode and cathode potential.

**Figure 7 sensors-19-04647-f007:**
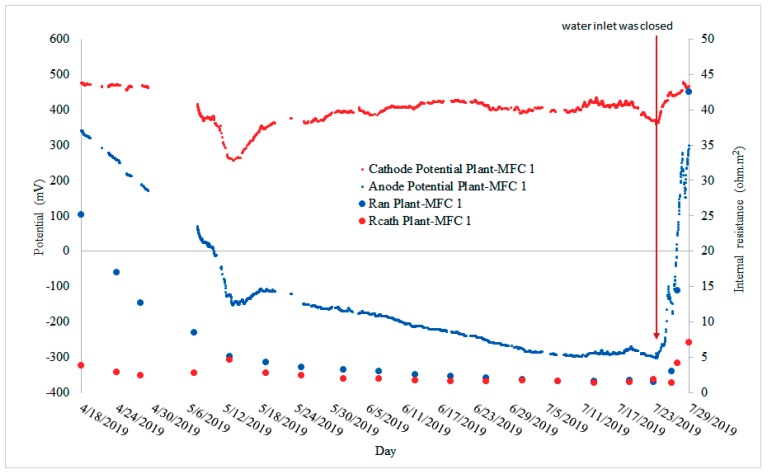
Anode and cathode internal resistance on Plant-MFC 1 during the fourth crop season.

**Figure 8 sensors-19-04647-f008:**
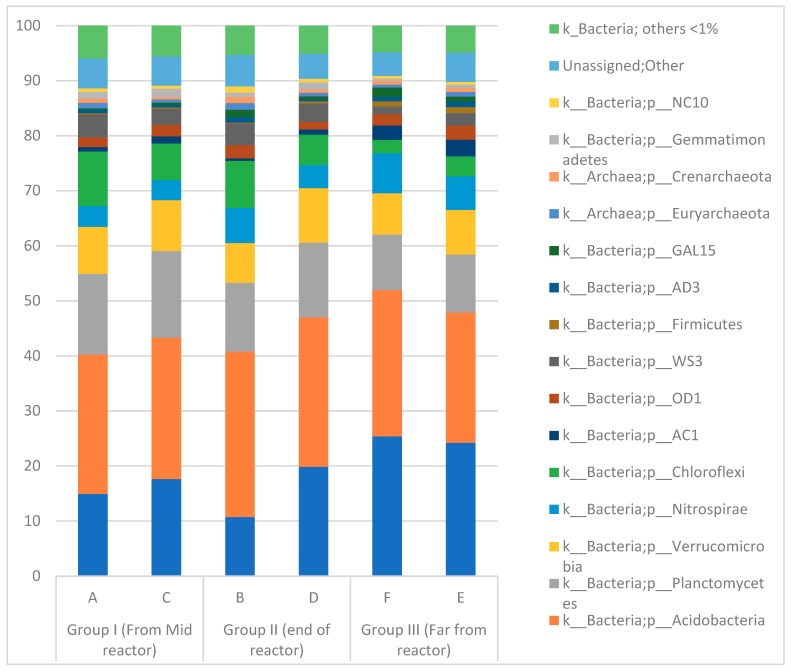
Microbial communities in the paddy field soil with relative abundance >1%.

**Figure 9 sensors-19-04647-f009:**
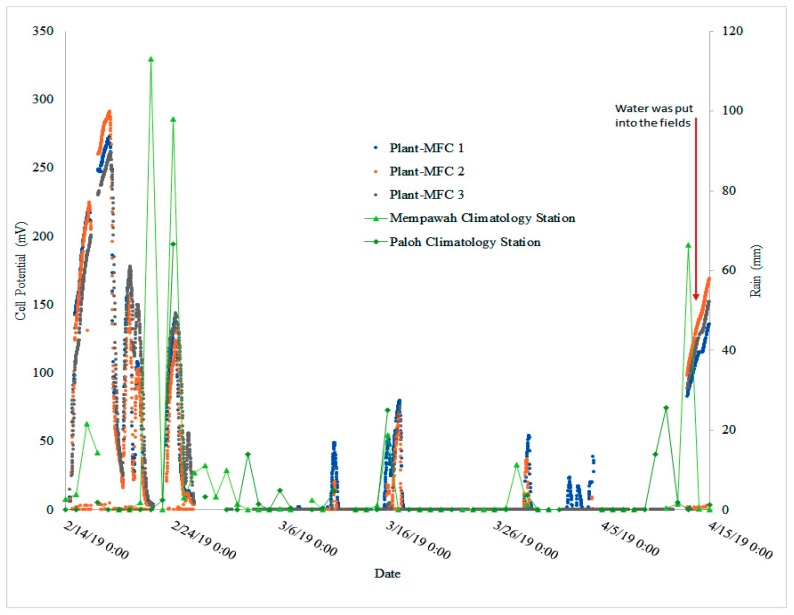
Cell potential peak (possibly) due to rain events. The rain events data were obtained from the two nearest climatology stations.

**Table 1 sensors-19-04647-t001:** Research phase on four different crop seasons.

Crop Season	Transplantation (Date)	Harvesting (Date)	Applied Load Plant-MFC (ohm)	Electricity Measurement
I	6 November 2017	10 February 2018	470	Multimeter
II	16 May 2018	6 August 2018	470	Multimeter
II	8 November 2018	13 February 2019	470	Multimeter
IV	13 May 2019	7 August 2019	1000	LORA

**Table 2 sensors-19-04647-t002:** Average aboveground wet biomass (kg/m^2^).

Location	3rd Crop Season	4th Crop Season
Above Plant-MFC	3 ± 0.5	4.6 ± 0.7
One meter from Plant-MFC	2.3 ± 0.5	3.4 ± 0.8

**Table 3 sensors-19-04647-t003:** Performances from selected Plant-MFC systems using rice of tubular designs.

Plant-MFC Types, Anode and Cathode Materials	Plant Species	Maximum Power Density (mW/m^2^ PGA)	Reference
Paddy field Anode: Graphite felt below the soil surface Graphite felt above the soil surface, air cathode	Rice (*Oryza sativa*)	6	[18]
Lab, Container Plant-MFC Anode: graphite mat and graphite rod Cathode: graphite granule, graphite rod, 100 mM K_3_Fe(CN)_6_	Rice (*Oryza sativa*)	30	[12]
Paddy field Anode: Graphite felt Cathode: Graphite felt modified with platinum catalyst, air cathode	Rice (*Oryza sativa*)	14	[22]
Paddy field Anode: graphite felt Cathode: graphite felt modified with platinum catalyst, air cathode	Rice (*Oryza sativa*)	19	[19]
Paddy field Anode: Graphite felt Cathode: Graphite felt with platinum catalyst modified with polystyrene-foam bars to maintain buoyancy	Rice (*Oryza sativa*)	80	[20]
Lab-Perspex tubes Anode: graphite granule, vermiculite, carbon rod. Cathode: graphite felt interwoven carbon felt, air cathode	Rice (*Oryza sativa*)	72	[6]
Rice paddy-field MFC Anode: Circular graphite felt Cathode: Graphite felt with platinum catalyst, air cathode	Rice (*Oryza sativa*)	140	[23]
PVC Pot Plant-MFC Anode: graphite felt interwoven with copper wire, soil from paddy field Cathode: graphite felt interwoven with copper wire, air cathode	Rice (*Oryza sativa*)	4.5	[41]
Lab-tubular Plant-MFC from PVC with membrane Anode: graphite felt, graphite granule, golden wire Cathode: thick graphite felt, golden wire current collector, 5 mol.m^−3^ K_3_Fe(CN)_6_	Reed mannagrass (*Glyceria maxima*)	18 12 *	[24]
Tubular Plant-MFC with membrane and silicone gas diffusion layer in a lab constructed wetland. Anode: graphite felt and graphite stick Cathode: graphite felt with golden wire current collector, air cathode	*Phragmites australis* *Spartine anglica*	22 ^a^ 82 ^a^	[17]
Tubular Plant-MFC without membrane with silicone tube air way in paddy field. Anode: graphite felt and titanium wire current collector Cathode: graphite felt and titanium wire current collector, air cathode	Rice (*Oryza sativa*)	44 9.6 ^a^ 8.5 ^b^	This study

* Average power density; ^a^ maximum daily average power generation; ^b^ average continuous power generation for 60 days.

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
