# Peer review of "Performance and Long Distance Data Acquisition via LoRa Technology of a Tubular Plant Microbial Fuel Cell Located in a Paddy Field in West Kalimantan, Indonesia"

_sensors, 2019, doi:10.3390/s19214647_

Round 1
Reviewer 1 Report
The manuscript presented a comprehensive studies and real situation operation of plant based microbial fuel cell device. The article is well articulated and authors have performed extensive long term stability study of the device in a rice paddy field. The article deserves publication in Sensors after addressing a few minor comments.
In the first paragraph of the introduction (line 35-42), giving an equation at cathode and anode of MFC to describe its working principle is recommended. Authors should comment on the effects of fertilizers and pesticides addition in the rice paddy field and how they affect the electron transfer from anode to cathode. The effect of water level in the field can change over extended period (because of rainfall or evaporation). How MFC device performance changes with this? In Fig. 7, a peak in the anode and cathode potential profile can be noted. Authors should comment on this. Different families of bacteria were found in the vicinity of MFC devices. Can authors conclude which family of bacteria is most suitable for the best device performances.
Author Response
Dear reviewer 1,
Thank you for your contribution to improve our manuscript. Here is our responses regarding to your comments.
In the first paragraph of the introduction (line 35-42), giving an equation at cathode and anode of MFC to describe its working principle is recommended.
Agree. We have added texts in lines 40-49:
The electricity production depends on the anode and the cathode potential differences, in which the cell potential is the cathode potential minus the anode potential[2–4]. The common approach to estimate the theoretical cell potential is using thermodynamics of the anode (e.g acetate) and the cathode (e.g oxygen) reaction which calculated based on Gibbs free energy at a specific condition [2,3]. For instance, the acetate oxidation (C2H3O2- + 4 H2O --> 2 HCO3- + 9 H+ + 8 e- ) and oxygen reduction (O2 + 4 H+ + 4 e- --> 2 H2O ) potential at a specific condition (acetate concentration 0.05M, [H2O] = 1M; pH = 7; pO = 0.2 bar; T = 298K) against Ag/AgCl reference electrode are -0.494 V and 0.6 V, respectively[3]. Therefore, a theoretical MFC cell potential using acetate model substrate and oxygen as the final electron acceptor is 1.094V.
Authors should comment on the effects of fertilizers and pesticides addition in the rice paddy field and how they affect the electron transfer from anode to cathode.
We have added an explanation on this in lines: 339-345
In addition, it could be that the degradation of Isoprocarb/Mipcin (from Mipcindo 50) and Imidacloprid (both chemicals have aromatic structure like in hexaclorobenzene) might provide an additional substrate for the EAB to generate electricity as have been shown that pesticide hexachlorobenzene degradation is enhanced in a soil MFC while generating electricity[37]. Also, it is known that addition of compost can affect current generation in Plant-MFC as well as that specific nutrients could include additional electron donor for electricity generation like in the case for urea obtained from urine[38–40]
The effect of water level in the field can change over extended period (because of rainfall or evaporation). How MFC device performance changes with this?
During the wet period, the water level was kept between 5-10 cm above the ground (section 2.1 Paddy field line 109) Fig. 1 illustrates how the water level was maintained over time. Therefore the effect of rainfall or evaporation during wet period is not observed since the soil/the Plant-MFC was kept anaerobic. In addition, we discuss the effect of the rainfall on section 3.4 (Plant-MFC can indicate a rain event as biosensor) during the dry period. However, we do agree that further precise monitoring on water level, precipitation and current over time is required.
In Fig. 7, a peak in the anode and cathode potential profile can be noted. Authors should comment on this.
We added short explanation on these peaks in line 366-368.
The weak peaks in the anode and the cathode potential profile in Fig.7 may be related to diurnal (day and night) effect as described in other studies [15]. However we did not specifically study this effect on this article.
Different families of bacteria were found in the vicinity of MFC devices. Can authors conclude which family of bacteria is most suitable for the best device performances?
Since no enrichment was studied by monitoring microbial dynamics over time we cannot reflect on these. Likely various unidentified microbes do exist that could be suitable for electricity generation; for this enrichment/purification and testing in MFC is e.g. required
We hope that we have provided sufficient information regarding your comments.
Reviewer 2 Report
This is a correct manuscript showing results of the investigations on plant MFC. The results are clearly presented and properly described. In my opinion the text is ready to be published.
in my opinion the article is acceptable for publication. The authors applied plant MFC in a paddy field what is an interesting trial of practical use of MFC technology. The results are presented clearly, the discussion is satisfactory to me. The authors very honestly compared their results with other works.Very interesting aspect of presented investigations is using plant MFC as a humidity sensor what can be helpful for resonable water usage in sustainable agriculture. The authors have also undertaken the effort of identifying microbial communities in paddy field soil. Though obtained power production in described plant MFC is low, it is comparable with similar works in this area. However, development of MFC technology towards its application in practice needs broad investigations on many aspects and for this reason I find this research important. As not-English native speaker I do not feel qualified to judge English language of the text.
Author Response
Dear reviewer 2,
We thank you for your suggestion to publish our paper.
Best regards,